# The Association between Nursing Skill Mix and Mortality for Adult Medical and Surgical Patients: Protocol for a Systematic Review

**DOI:** 10.3390/ijerph17228604

**Published:** 2020-11-19

**Authors:** Diana Kushemererwa, Jenny Davis, Nompilo Moyo, Sue Gilbert, Richard Gray

**Affiliations:** School of Nursing and Midwifery, La Trobe University, Melbourne, Bundoora 3086, Victoria, Australia; J.Davis@latrobe.edu.au (J.D.); n.moyo@latrobe.edu.au (N.M.); s.gilbert@latrobe.edu.au (S.G.); r.gray@latrobe.edu.au (R.G.)

**Keywords:** inpatient, skill mix, nurse-to-patient ratio, mortality, observational research, systematic review, protocol

## Abstract

Skill mix refers to the number and educational experience of nurses working in clinical settings. Authors have used several measures to determine the skill mix, which includes nurse-to-patient ratio and the proportion of baccalaureate-prepared nurses. Observational studies have tested the association between nursing skill mix and patient outcomes (mortality). To date, this body of research has not been subject to systematic review or meta-analysis. The aim of this study is to systematically review and meta-analyse observational and experimental research that tests the association between nursing skill mix and patient mortality in medical and surgical settings. We will search four key electronic databases—MEDLINE [OVID], EMBASE [OVID], CINAHL [EBSCOhost], and ProQuest Central (five databases)—from inception. Title, abstract, and full-text screening will be undertaken independently by at least two researchers using COVIDENCE review management software. We will include studies where the authors report an association between nursing skill mix and outcomes in adult medical and surgical inpatients. Extracted data from included studies will consist measures of nursing skill mix and inpatient mortality outcomes. A meta-analysis will be undertaken if there are at least two studies with similar designs, exposures, and outcomes. The findings will inform future research and workforce planning in health systems internationally.

## 1. Introduction

### 1.1. Nursing Skill Mix 

The World Health Organization (WHO) states that determining the right mix of health personnel—physicians, nurses, and allied health professionals—is a longstanding challenge for health care organizations and health systems [1]. Health care is a labour-intensive activity. Approximately 62% of the direct costs of running a hospital relate to staff costs according to the Australian Institute of Health and Welfare [2]. Ensuring that a health system has optimal staffing in terms of numbers and skills is key to optimising health outcomes for patients [3]. There has been increased policy attention paid to the impact of nursing skill mix on health care delivery [4]. 

Despite the reported need for more robust guidelines based on a sound evaluation of existing skill mix patterns and better dissemination of good practices to expand the evidence base and support informed decision making in this area [3], the association between optimal numbers of staff and their skill mix remains unknown [5]. 

### 1.2. Definition of Skill Mix

Nursing skill mix is a commonly used term in health policy and practice. In broad terms, skill mix refers to the number, educational preparation of nurses or level of nurses, and their experience working in a clinical setting (typically a hospital). Nurse staffing means that an appropriate number of nurses is always available across the continuum of care, with a proper mix of education, skills, and experience to ensure that patient care needs are met [6]. 

Different measures of skill mix focus on different aspects of nurse staffing, which include nurse-to-patient ratio [7,8,9,10], qualified-to-unqualified ratio [11], nursing care hours per patient day [12], nursing hours available per patient day [13,14,15,16,17], grade mix [18], length of clinical experience [19,20], and educational preparation [20,21,22,23,24]. The nurse-to-patient ratio (NPR) is probably the most used measure of skill mix. It describes the estimated number of patients to whom an individual nurse provides care for during a defined period (typically a shift) [9]. This definition assumes the equal distribution of workload among nurses and omits other care staff (e.g., nursing assistants) working in the ward during the same shift [8,25]. An alternative method of calculating the nurse-to-patient ratio is nursing care hours per patient day (NHPPD), which is an estimate of the maximum number of nursing care hours available to patients each day. Notably, NHPPD is not a measure of the actual amount of direct nursing care a patient received. Instead, it is an estimate of the absolute number of care hours available on a given day [26].

The clinical grade—a proxy measure of seniority and competence—of nurses providing care was proposed as a further dimension of skill mix [27]. Internationally, countries tend to have bespoke grading systems. For example, in England, there is a banding system ranging from five (staff nurse) through eight (nurse consultant) [27]. In Australia, a nurse’s classification and related definitions are linked to Australia’s industrial relations agreements and enterprise bargaining agreements (EBAs) and approved nationally by Fair Work Australia under the Australian Fair Work Act [28]. Nurse classifications are subject to change during negotiation cycles and vary across different Australian states and territory jurisdictions. In the Australian state of Victoria, for example, the grading system ranges from level 1 to 5 [29]. The clinical experience of nurses is a related, although discreet, measure of skill mix. Nurses of higher clinical grade are generally considered to be more clinically experienced [27]. The experience is typically calculated as the number of years since first registration [27]. This measure likely overestimates actual clinical experience because nurses will inevitably have had career breaks (e.g., parental leave), especially in a woman-dominated profession [30].

The educational preparation of nurses is seen as an important measure of skill mix. Authors tend to report the level of educational preparation of nurses as the percentage of nurses working in a clinical setting that hold at least a baccalaureate degree [9,20,21,22,31]. In this review, we define nurses as the percentage of Registered Nurses (RNs) in a clinical setting. Different countries have set targets for the proportion of the workforce that should have a degree. For example, in the USA, the Institute of Medicine (IoM) has stipulated that 80% of nurses should hold a baccalaureate qualification [32]. 

Hospital wards in many countries are staffed by a mixture of qualified (licensed or registered) nurses and unqualified nursing (or health care) assistants [33]. De Cordova et al. [34] argued that an important yet overlooked aspect of skill mix is the ratio of registered (or licensed) nurses to unregistered care assistants working a shift. This is a more precise measure of the total number of staff providing care and is considered a proxy of the care team’s expertise. 

In summary, skill mix is a multi-component construct that seeks to capture the number, experience, and education of nurses or levels working in healthcare settings. Several high-profile studies have examined the association between different aspects of nursing skill mix and patient outcomes. Twigg et al. [35] reported the only systematic review of observation research relevant to this review. They tested the association between nursing hours per patient day and a range of nursing-sensitive patient outcomes (e.g., length of stay, failure to rescue, and mortality). Twigg et al. [35] reported a narrative analysis of the included studies. Of the 95 observational studies included in their systematic review, the narrative summary comprised 63 reports. A meta-analysis was not reported because of variations in how the exposure (skill mix) and outcomes (e.g., length of stay and mortality) were determined. The authors concluded that of those studies identifying significant results, there were 12 outcomes in which three-quarters or more of those studies found an inverse association with increases in nurse skill mix: length of stay, ulcer, gastritis and upper gastrointestinal bleeds, acute myocardial infarction, restraint use, failure-to-rescue, pneumonia, sepsis, urinary tract infection, mortality/30-day mortality, pressure injury, infections (excluding urinary tract infections), and shock/cardiac arrest/heart failure. The authors determined that there was insufficient evidence to draw inferences about causal relationships since research into nursing-sensitive outcomes continues to suffer methodological flaws and heterogeneity of results. 

There are critical methodological limitations with this review, including a lack of clarity of how multiple reports from the same study were handled. The Cochrane handbook [36] offers guidance on how to do this (i.e., reports from a single study need to be linked together, so it is transparent to the reader); seemingly, the authors did not do this. For example, two studies [21,37] used data from the same data set (RN4CAST) but are listed (online supplementary document C3) as separate studies in the review [35]. Study quality was appraised using the Joanna Briggs Institute (JBI) critical appraisal tool. However, if authors only refer to JBI tools generally, it is unclear how many tools are being employed [38]. A summary of critical appraisal results for included studies using the JBI risk of bias is only reported as supplementary data and is not described in the main manuscript. 

Given the methodological limitations in the previous systematic review, there is a need for a robust systematic review and meta-analysis to test the association between nursing skill mix and patient outcomes. Patient outcomes significantly impact nursing care quality, the cost of adequate care, and healthcare policy-making formulation [39].

## 2. Materials and Methods 

### 2.1. Research Objectives

The objective is to systematically review, appraise, and meta-analyse the evidence from observational and experimental studies reporting the association between nursing skill mix and mortality for adult medical and surgical patients.

### 2.2. Review Question

What is the association between nursing skill mix and patient mortality for adult medical and surgical patients?

### 2.3. Design

We will consider published studies that addressed the research objective. We included observational (e.g., cohort, case-control) and experimental (e.g., randomised controlled trials) studies. Our protocol complied with the Preferred Reporting Items for Systematic Reviews and Meta-Analyses Protocols checklist 2015 (PRISMA-P) [40]. Our review protocol was prospectively registered with PROSPERO (CRD42019139124). 

### 2.4. Eligibility Criteria 

We will include studies meeting the following inclusion criteria:Adult medical and surgical inpatients aged at least 18 years of age;The exposure or intervention is a nursing skill mix that includes, but is not limited to, nurse-to-patient ratio, nursing hours per patient per day, and nurse education level;Report patient mortality (any definition) as an outcome;Published in English.

We will exclude qualitative studies and studies that included emergency departments, maternity (women having a baby), and psychiatric/mental health. We decided not to include studies that focused on clinical settings other than medical and surgical because of the potential for high levels of heterogeneity. For example, nursing care in mental health settings likely differs considerably from that in general wards.

### 2.5. Searching the Grey Literature

We did not intend to search the grey literature for the following reasons: it is difficult to describe a replicable search strategy, studies can be difficult to retrieve, reports and papers are not necessarily permanently archived on websites, and studies may not have been subject to peer review. 

### 2.6. Search Strategy

Our search strategy was developed to address three key concepts: hospital, nursing skill mix, and mortality. We intend to search the following databases: MEDLINE [OVID], EMBASE [OVID], CINAHL [EBSCOhost], and ProQuest Central (5 databases). No restrictions will be placed on the date of publication because we aim to capture all relevant literature and there were no specific dates on which the first skill mix studies were published that might enable us to restrict the time frame of our research. Our initial search strategy was developed in MEDLINE (Table 1) and adapted for each subsequent database. 

### 2.7. Study Selection

Studies identified in the database search will be exported to Endnote. The reference management file will then be exported to COVIDENCE (a web-based software platform for managing systematic reviews) for the title, abstract, and full-text screening. Duplicate records will be excluded. The titles and abstracts of all articles will be screened by two researchers independently for eligibility against the review inclusion criteria. A third reviewer will resolve any disagreements. Full texts of papers will then be uploaded to COVIDENCE and reviewed by two researchers; again, disagreements will be resolved by a third member of the research team. A Preferred Reporting Items for Systematic Reviews and Meta-Analyses (PRISMA) flow chart will be produced to show the flow of studies through the selection process.

### 2.8. Managing Multiple Reports from the Same Study

Included studies may be based on large datasets, and multiple papers will likely be reported from each dataset. There is a risk that we may include two (or more) studies that used a single data source, potentially distorting the results of our review and meta-analysis. To avoid this, we will adhere to the procedures for managing multiple outputs from the same study described in the Cochrane handbook [36]: 1. We will check to see if any studies used the same study acronym, 2. the sample sizes of studies will be checked to see if any studies have identical sample sizes, 3. dates and locations of fieldwork will be compared, and 4. study authors will be compared to determine if they are similar. Based on the information extracted from the papers, we will decide as to whether the papers are from a single study. If any ambiguity remains, we will write to the authors to confirm that the studies were from the same dataset. Where we have two or more papers from the same dataset, we will only include the main published report from the identified dataset (defined as the first paper published). Where there is redundant or duplicate reporting between papers, we will write to journal editors to inform them of our concerns. Multiple reports of the same study will be collated into summary tables that will be reported as appendices to our manuscript to ensure that each study, rather than each report, is included in the review [36].

### 2.9. Data Extraction

Data from eligible studies will be extracted using a data extraction tool developed for this review based on common items in systematic reviews. Pilot testing will be conducted to ensure that all relevant data are extracted from included studies. The following data will be extracted: citation (author, and the year when the paper was received, accepted, and published), country or region where fieldwork was undertaken, the number of participating hospitals, study registration status, design, participants (the number of patients, the number of nurses), sample size, sampling frame, exposure, the measure of nursing skill mix, how was the outcome determined (e.g., in-hospital mortality, 30-day mortality), and results (i.e., findings reported against each outcome). Two reviewers will perform data extraction independently, with disagreements resolved by discussion with a third reviewer. 

### 2.10. Risk of Bias

The risk of bias will be determined using the Effective Public Health Practice Project (EPHPP) measure [41]. The Effective Public Health Practice Project tool evaluates the risk of bias across six components: study design, confounding, blinding, data collection methods, withdrawals, and dropouts, rated on a three-point scale (good, fair, and poor). Based on the component rating, the reviewer then makes an overall rating on a different three-point scale (strong, moderate, and weak). Two reviewers will independently complete the risk of bias assessment, discuss any discrepancies in component ratings, and report a reason for the discrepancy. If a disagreement occurs between reviewers, a third reviewer will be consulted. Studies with a high risk of bias will not be excluded, but a sensitivity analysis will be conducted (i.e., we will compare the strength of the association between low and high-risk studies). We will produce a table that summarises the risk of bias against each component and for the study overall. 

### 2.11. Assessing Clinical and Methodological Heterogeneity

Both the clinical (variability in participants, interventions, and outcomes) and methodological heterogeneity (study design, outcome measurement tools, and risk of bias) of the included studies will be determined by manually reviewing included studies based on the primary data extraction table. We will then need to decide whether we are able to pool two or more studies together, or if studies differ substantially, they will need to be analysed as subgroups. A supplementary file for the pooled analyses will be provided.

### 2.12. Statistical Heterogeneity

Statistical heterogeneity refers to differences in the effects of the exposure on study outcomes. The heterogeneity between included studies will be determined by visual inspection of forest plots and the calculation of the *I*^2^ statistic, which provides a numerical estimate of heterogeneity [42]. 

The *I*^2^ statistic will be calculated using RevMan [43]. The *I*^2^ statistic describes the percentage of the total variation across studies that is due to heterogeneity rather than chance, and results lie between 0% and 100%. The level of heterogeneity increases with an increase in value. Conventionally, a value of 0% indicates no heterogeneity, 25% low, 50% moderate, and 75% high heterogeneity [44]. 

If *I*^2^ is between 50% and 75%, we will undertake a meta-analysis but include a caveat to let the reader know that we had concerns about the reliability of our results. For *I*^2^ > 75%, the results will not be pooled [44]. 

### 2.13. Meta-Analyses

Outcome data will be pooled using a random-effects model. Review Manager software (RevMan, version 5.3, Copenhagen, The Nordic Cochrane Centre, The Cochrane Collaboration, 2017) [45] will be used to complete data extraction and meta-analyses. Where there are at least two studies that report similar designs, exposure, and outcomes measures, data will be combined by meta-analysis to calculate pooled effect estimates and their 95% confidence intervals using a random-effects model. 

### 2.14. Subgroup Analyses (Planned)

Subgroup analyses will be used to determine whether different effects are observed in the different subgroups of participants. We anticipate planned subgroup analyses based on the skill mix measure used in the studies (e.g., nurse-to-patient ratio, nursing hours per patient day, and nurse education level). We will perform a meta-analysis for each subgroup to determine the overall effect in comparison to different groups.

### 2.15. Sensitivity Analysis

A sensitivity analysis is a repeat of the primary analysis or meta-analysis in which alternative decisions or ranges of values are substituted for arbitrary or unclear decisions. For example, suppose some studies’ eligibility in the meta-analysis is obscure because they do not contain full details. In that case, sensitivity analysis may involve undertaking the meta-analysis twice: the first time, including all studies, and second, including only those known to be eligible. We primarily intend to use sensitivity analysis to assess the results’ robustness by contrasting the methodological quality of included studies (i.e., low and high risk of bias). 

## 3. Discussion

We have described a detailed protocol for a rigorous systematic review and meta-analysis to address an identified gap in the evidence base. Each step of the review will be conducted according to the current best practices. Two reviewers will undertake each step of the review process. A third reviewer will resolve any disagreements. 

### Study Limitations

We will include studies from both medical and surgical settings. It could be argued that we should adopt a specific focus on either surgical or medical settings. This would ensure that included study populations are similar in terms of the nursing care that patients received.

We will exclude studies where at least some of the participants were under the age of 18 years. Potentially, we may omit studies that have relevant data. However, it will not be possible to extract these data and, consequently, we have decided to exclude them from our study.

Studies conducted in mental health settings will also be excluded because psychiatric nursing care is likely different from that provided in general medical and surgical settings. We note however that a review of mental health skill mix research would be informative.

## 4. Conclusions

Our review’s findings will provide a robust summary of the state of knowledge about the association between nursing skill mix and patient mortality.

## Figures and Tables

**Table 1 ijerph-17-08604-t001:** Medline search.

Search ID	Search Terms	Search Notes
		Concept 1: Context or population (hospital inpatient)
S1	Exp hospital/	
S2	Exp hospitalization/	
S3	Exp Female/or exp Male/or exp Adult/	
S4	Exp hospital patient/	
S5	(hospital or hospitali?ation or inpatient* or hospital patient)	
S6	S1 or S2 or S3 or S4	
		Concept 2: Nursing skill mix terminology
S7	Exp skill mix/	
S8	Exp nursing staff/	
S9	Grade mix or hours per patient day or hppd or nurse-patient ratio or nurs* staff or nurs*staff mix or nursing grades or registered nurs* or unregistered nurs* or personnel staffing or licensed nurs* or trained nurs* or staffing levels or clinical competence	
S10	Exp nurse-patient ratio	
S11	S7 or S8 or S9 or S10	
		Concept 3: Mortality outcome
S12	Exp hospital mortality/	
S13	S6 AND S10 AND S12

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
