# Peer review of "The Association between Nursing Skill Mix and Mortality for Adult Medical and Surgical Patients: Protocol for a Systematic Review"

_ijerph, 2020, doi:10.3390/ijerph17228604_

Round 1

Reviewer 1 Report

Dear authors
Thank you for the opportunity to review:  "The Association between Nursing skill mix and 2 Clinical Outcomes for Adult Medical and Surgical 3 Patients: Protocol for a systematic review"

This study is well organized with a significant contribution to the field of Nursing. Introduction provides a significant bibliography for this study.

The objective is appropriate and  guide the study correctly. It is well organized and comprehensively described.

Finally, the bibliographic references should be reviewed, for example number 46.

This paper provide a summary of the state of knowledge about the association between nursing skill mix and patient mortality. This study could be useful in the future research.

Author Response

Our response: number 46 has changed to 44. We have reviewed and amended our referencing.

[44] M. Borenstein, Hedges, V Larry, Higgins, P. T Julian, Rothstein, R.Hannah, Ebook Library, & ProQuest. (2009). Introduction to Meta-Analysis (2nd.ed.). Hoboken: Wiley

Reviewer 2 Report

Overall,  a very interesting and well-written manuscript

lines 90 and 93: should use the same tense, eg, report or reported

in lines, 82,  90 and 93 also the authors should have started with the name and then the number, eg. "according to Twing et al [38]..."

in line 108 better replace the word "papers" with "studies"

in lines 156-157 the authors did not justify why "No restrictions will be placed on the date of publication"

in lines 190 -191, "the data extraction tool developed explicitly for this review" need to be supported by the literature, or to be tested previously, or to be justified

references 3,4,6 very old. I believe the introduction should be supported from more updated studie 

Author Response

Reviewer comment: lines 90 and 93: should use the same tense, e.g., report or reported

Our response: Tense changed to reported and sentence started with Twigg et al. [37]

Reviewer comment: in lines, 82, 90 and 93 also the authors should have started with the name and then the number, e.g. "according to Twigg et al [38]..."

Our response: We have amended the text accordingly. Line 83 is now; De Cordova et al. [37], Line 91, and 94 start with Twigg et al. [37]

Reviewer comment: in line 108 better replace the word "papers" with "studies"

Our response: We have amended the text accordingly.

Reviewer comment: in lines 156-157 the authors did not justify why "No restrictions will be placed on the date of publication"

Our response: We have amended the manuscript accordingly. 159- 161

No restrictions will be placed on the date of publication because we sought to capture all relevant literature and there are no specific dates when the first skill mix studies were published that might enable us to restrict the time frame of our research.

Reviewer comment: in lines 190 -191, "…the data extraction tool developed explicitly for this review" need to be supported by the literature, or to be tested previously, or to be justified

Our response: We have amended the manuscript accordingly. 194-196

Data from eligible studies will be extracted using a data extraction tool developed for this review based on common items in systematic reviews. Pilot testing will be conducted to ensure that all relevant data are extracted from included studies.

Reviewer comment: references 3,4,6 very old. I believe the introduction should be supported from more updated studies 

Response 6: We have updated references 3, 4,6 accordingly.

[3] C. Dubois, & D. Singh, From staff-mix to skill-mix and beyond: Towards a systemic approach to Health workforce Management. Hum. Resour. Health 2009, Vol 7. doi:10.1186/1478-4491-7-87.

[4]   J. Needleman, Nursing skill mix and patient outcomes. BMJ Quality & Safety 2016 , doi:10.1136/bmjqs-2016-006197.

[5]   National Institute for Health Research. (2019). Staffing on wards: Making decisions about healthcare staffing, improving effectiveness and supporting staff to care well.

Reviewer 3 Report

Thank you for giving me the opportunity to review this interesting and well-written paper. 

I have few suggestions:

I would change the title according to the fact that you will analyze mortality and not other clinical outcomes.

If you want to study the association between skill mix and mortality, I suggest to not include survey but only cohort, case control and experimental studies.

Please define better why you decide to include only medical and surgical areas.

I suggest to screen also the references of the papers included in the systematic review.

Author Response

Reviewer comment: I would change the title according to the fact that you will analyze mortality and not other clinical outcomes. 

Our response: We have amended the title of the manuscript to;

The Association between Nursing skill mix and mortality for Adult Medical and Surgical Patients: Protocol for a systematic review

Reviewer comment: If you want to study the association between skill mix and mortality, I suggest to not include survey but only cohort, case control and experimental studies.

Our response: line 131-133 surveys excluded. We will include observational (e.g., cohort, case-control) and experimental (e.g., randomised controlled trials) studies.

Reviewer comment: Please define better why you decide to include only medical and surgical areas.

Our response: line 146-148; We decided not to include studies that focused on clinical settings other than medical and surgical because of the potential for high levels of heterogeneity. For example, nursing care in mental health settings likely differs considerably to that in general wards.

Reviewer comment: I suggest to screen also the references of the papers included in the systematic review.

Our response: All references of papers included in the systematic review were screened

Reviewer 4 Report

The abstract requires revisión: the objective of the study is not clear.

A section on strengths and limitations would be advisable to replace the discussion and conclusion sections.

References should include a greater number of studies published in the last five years

Author Response

Reviewer comment: The abstract requires revision: the objective of the study is not clear.

Our response: The aim of this study is to systematically review and meta-analyse observational and experimental research that tests the association between nursing skill mix and patient mortality in medical and surgical settings.

Reviewer comment: A section on strengths and limitations would be advisable to replace the discussion and conclusion sections.

Our response: We have included the limitations. Line 269-277

We will include studies from both medical and surgical settings. It could be argued that we should adopt a specific focus on either surgical or medical settings. This would ensure that included study populations were similar in terms of the nursing care that patients received. 

We will exclude studies where at least some of the participants are under the age of 18 years. Potentially we may omit studies that have relevant data. However, it will not be possible to extract these data and consequently, we have decided to exclude them from our study.

Studies conducted in mental health settings will also be excluded because psychiatric nursing care is likely different from that provided in general medical and surgical settings. We note however that a review of mental health skill mix research would be informative.

Reviewer comment: References should include a greater number of studies published in the last five years

Our response: We have amended the references accordingly.